# 3-Bromo-4,5-dihydroxybenzaldehyde Protects Keratinocytes from Particulate Matter 2.5-Induced Damages

**DOI:** 10.3390/antiox12061307

**Published:** 2023-06-20

**Authors:** Ao-Xuan Zhen, Mei-Jing Piao, Kyoung-Ah Kang, Pincha-Devage-Sameera-Madushan Fernando, Herath-Mudiyanselage-Udari-Lakmini Herath, Suk-Ju Cho, Jin-Won Hyun

**Affiliations:** 1Department of Biochemistry, College of Medicine, Jeju National University, Jeju 63243, Republic of Korea; zhenaoxuan705@stu.jejunu.ac.kr (A.-X.Z.); sameera@stu.jejunu.ac.kr (P.-D.-S.-M.F.); lakmini@stu.jejunu.ac.kr (H.-M.-U.-L.H.); 2Jeju Research Center for Natural Medicine, Jeju National University, Jeju 63243, Republic of Korea; mjpiao@jejunu.ac.kr (M.-J.P.); legna07@jejunu.ac.kr (K.-A.K.); 3Department of Anesthesiology, Jeju National University Hospital, College of Medicine, Jeju National University, Jeju 63241, Republic of Korea

**Keywords:** particulate matter 2.5, 3-bromo-4,5-dihydroxybenzaldehyde, reactive oxygen species, skin damage

## Abstract

Cellular senescence can be activated by several stimuli, including ultraviolet radiation and air pollutants. This study aimed to evaluate the protective effect of marine algae compound 3-bromo-4,5-dihydroxybenzaldehyde (3-BDB) on particulate matter 2.5 (PM_2.5_)-induced skin cell damage in vitro and in vivo. The human HaCaT keratinocyte was pre-treated with 3-BDB and then with PM_2.5_. PM_2.5_-induced reactive oxygen species (ROS) generation, lipid peroxidation, mitochondrial dysfunction, DNA damage, cell cycle arrest, apoptotic protein expression, and cellular senescence were measured using confocal microscopy, flow cytometry, and Western blot. The present study exhibited PM_2.5-_generated ROS, DNA damage, inflammation, and senescence. However, 3-BDB ameliorated PM_2.5_-induced ROS generation, mitochondria dysfunction, and DNA damage. Furthermore, 3-BDB reversed the PM_2.5_-induced cell cycle arrest and apoptosis, reduced cellular inflammation, and mitigated cellular senescence in vitro and in vivo. Moreover, the mitogen-activated protein kinase signaling pathway and activator protein 1 activated by PM_2.5_ were inhibited by 3-BDB. Thus, 3-BDB suppressed skin damage induced by PM_2.5_.

## 1. Introduction

Fine particulate matter 2.5 (PM_2.5_) causes air pollution from various sources, such as coal burning, transport, and anthropogenic emissions [1]. Approximately 90% of human beings face health risks from pollution, which violates the WHO Air Quality Guidelines [2]. PM_2.5_ induces damage in vitro and in vivo to the bronchial epithelium, human endothelial cells, and macrophage-like cells [3,4,5,6,7,8,9]. The effects of air pollutants on the human skin have become a global concern recently [10]. Moreover, skin directly exposed to PM_2.5_ can result in acute and chronic reactions. Recently, many studies, including ours, have outlined the potential mechanism by which PM_2.5_ triggers excessive formation of reactive oxidative species (ROS), leading to skin inflammation and senescence [11,12,13,14,15]. PM_2.5_ induces ROS generation, inflammatory cytokines, and apoptosis, and it promotes skin aging by interacting with p53, nuclear factor kappa B (NF-κB), interleukin-1 beta (IL-1β), IL-6, and caspase-3 [14,16,17].

3-Bromo-4,5-dihydroxybenzaldehyde (3-BDB), a natural marine compound from red algae (*Rhodomela confervoides*, *Polysiphonia morrowii*, and *Polysiphonia urceolata*), is known to have free radical scavenging, anticancer, and antibacterial properties [18,19,20]. We previously demonstrated that 3-BDB exerted antioxidant effects in keratinocytes by regulating nuclear factor and erythroid 2-like 2 (Nrf2) pathways. It also protects skin cells from ultraviolet B by inhibiting the generation of ROS [21,22,23]. Moreover, it inhibits macrophage infiltration, thereby improving cardiac function, preventing myocardial ischemia, and suppressing allergic inflammation [24,25,26]. However, little is known about the effects of 3-BDB on skin damage (senescence and apoptosis) caused by PM_2.5_.

Therefore, we aimed to elucidate the effect of 3-BDB on PM_2.5_-induced ROS generation, macromolecular damage, apoptosis, and senescence of skin cells in vitro and in vivo.

## 2. Materials and Methods

### 2.1. Sample Preparation

3-Bromo-4,5-dihydroxybenzaldehyde (3-BDB) was obtained from Matrix Scientific (Columbia, SC, USA). PM_2.5_ (NIST particulate matter SRM 1650b) was purchased from Sigma-Aldrich Co., Ltd. (St. Louis, MO, USA). 3-BDB and PM_2.5_ were dissolved in dimethyl sulfoxide (DMSO), and the DMSO concentration in the cell medium during treatment was maintained at <0.1%.

### 2.2. Cell Culture

The human HaCaT keratinocyte cell line was provided by Cell Lines Service (Heidelberg, Germany). They were cultured in Dulbecco’s modified Eagle’s medium (Life Technologies Co., Ltd., Grand Island, NY, USA), containing 10% heat-inactivated fetal calf serum (Life Technologies Co., Ltd.), and 1% antibiotic-antimycotic (Life Technologies Co., Ltd.) in a 37 °C incubator with a humidified atmosphere containing 5% CO_2_.

### 2.3. Animal Experiment

We used HR-1 hairless male mice (OrientBio, Seongnam, Republic of Korea) for in vivo experiments following guidelines of the Jeju National University (Jeju, Republic of Korea) (permit number: 2017-0026). Moreover, mice were divided into four groups (n = 4 per group): phosphate buffered saline, PM_2.5_ (100 µg/mL), 3-BDB (0.3 mM) + PM_2.5_, and 3-BDB (3 mM) + PM_2.5_. The dorsal portion of the skin of the mice was exposed to 3-BDB (0.3 mM or 3 mM) for 30 min before exposing them to PM_2.5_. Then, they were covered with the nonwoven polyethylene pad (over a 1 cm^2^ area), which dispersed PM_2.5_ daily for 7 consecutive days. Finally, on day 7, the skin tissues were dissected for Western blot analysis [12].

### 2.4. ROS Scavenging Ability

We used 2′,7′-dichlorodihydrofluorescein diacetate (H_2_DCFDA; Molecular Probes, Eugene, OR, USA) to measure the inhibition of PM_2.5_-induced ROS by 3-BDB. Cells (1.0 × 10^5^ cells/mL) were seeded into a 6-well plate. Cells were added to 10, 20, and 30 μΜ of 3-BDB or 1 mM of N-acetyl cysteine (NAC) for 1 h and then exposed to 50 μg/mL of PM_2.5_ for 30 min. Cells were stained with H_2_DCFDA (25 μΜ), and stained cells were detected using a FACSCalibur flow cytometer (Becton Dickinson, Mountain View, CA, USA). Similarly, cells were seeded into the chamber slides, and 30 μΜ of 3-BDB were treated for 1 h and then treated with PM_2.5_ (50 μg/mL) for 30 min. Cells stained with H_2_DCFDA were observed using an FV1200 laser scanning confocal microscope (Olympus, Tokyo, Japan).

### 2.5. Lipid Peroxidation Assay

We detected the suppression of PM_2.5_-induced oxidation of lipids by 3-BDB using a diphenyl-1-pyrenylphosphine probe (DPPP, 2 µM; Molecular Probes). Cells were seeded into chamber slides, treated with 30 μΜ of 3-BDB for 1 h, and exposed to 50 μg/mL of PM_2.5_ for another 24 h. Lipid peroxidation fluorescence was detected using a confocal microscope after DPPP staining.

### 2.6. Analysis of Mitochondria Function

We explored the mitochondrial calcium level and cell potential to access the inhibitory effect of 3-BDB on PM_2.5_-induced mitochondrial dysfunction. For mitochondrial calcium detection, cells were treated with 30 μΜ of 3-BDB for 1 h and exposed to 50 μg/mL of PM_2.5_ for another 24 h. The harvested cells were stained with Rhod-2 acetoxymethyl ester (Rhod-2 AM, 5 µM; Molecular Probes) and subjected to flow cytometry. We harvested cells stained with 5,5′,6,6′-tetrachloro-1,1′,3,3′-tetraethylbenzimidazolylcarbocyanine iodide (JC-1, 2 µM; Invitrogen, Carlsbad, CA, USA) to detect the mitochondrial membrane potential, and we captured the fluorescence using a flow cytometer or confocal microscope.

### 2.7. Detection of 8-Oxoguanine (8-OxoG)

8-OxoG is the most significant biomarker for oxidative DNA damage [27]. To detect 8-oxoG levels, we used avidin-tetramethylrhodamine isothiocyanate (TRITC) conjugate fluorescent dye (Sigma-Aldrich Co., Ltd.), which has an affinity to 8-oxoG [28]. Harvest cells in the chamber slide were treated with 30 μΜ of 3-BDB for 1 h and 50 μg/mL of PM_2.5_ for another 24 h. Then, cells were stained with avidin-TRITC conjugate, and their fluorescence intensity was estimated using a 1.8.0 software program of image J under the confocal microscope [12].

### 2.8. Comet Assay

We performed a comet assay to assess the effect of 3-BDB on PM_2.5_-induced DNA strand breaks. Cells (0.8 × 10^5^ cells/mL) were seeded into the microtubes and treated with 3-BDB and/or PM_2.5_ for 30 min. Harvested cells were fixed on the slides with 0.7% of agarose gel, immersed in lysis buffer (2.5 M NaCl, 100 mM Na_2_-EDTA, 10 mM Tris, and 1% N-lauroylsarcosinate, pH 10) for 1 h, electrophoresed for 20 min, and then dried. Images of total fluorescence and the change in DNA tail length were recorded using ethidium bromide (10 µg/mL) under a fluorescence microscope equipped with Komet 5.5 software program of image analysis (Kinetic Imaging, Liverpool, UK). Fifty cells were counted per slide.

### 2.9. Detection of IL-1β and IL-6

The IL-1β and IL-6 concentrations in the culture medium were measured using a human Quantikine ELISA kit (R&D Systems, Minneapolis, MN, USA). Cells were treated with 30 μΜ of 3-BDB for 1 h. Then, they were incubated for 24 h with 50 μg/mL of PM_2.5_ and then centrifuged at 3000 rpm for 15 min in the culture media. Cell-free supernatant was added to a 96-well plate coated with the primary antibodies against IL-1β and IL-6. The HRP-conjugated detection antibodies (100 µL) were then added and incubated for 1 h at 37 °C. After washing three times, the substrates were incubated for another 20 min. Finally, the stop solution was added to each well, and the absorbance of concentrations was measured at 450 nm using a SpectraMax i3x microplate reader (Molecular Devices, San Jose, CA, USA), which was performed immediately.

### 2.10. Western Blot

Cells were treated with 30 μΜ of 3-BDB for 1 h and then with 50 μg/mL PM_2.5_ for 24 h, and mice skin tissues were treated with 3-BDB and PM_2.5_ according to the above animal experiment method. Protein lysis buffers from the cells and mouse skin were loaded into a separating gel containing SDS-PAGE electrophoresis buffer. The target proteins were transferred onto membranes and shaken with primary and secondary antibodies sequentially. Finally, protein bands were obtained using the Amersham enhanced chemiluminescence, plus a Western blotting detection system (GE Healthcare, Buckinghamshire, UK). The primary antibodies used were as follows: actin (Sigma-Aldrich Co., Ltd.), c-Jun N-terminal kinase (JNK), p38 (Genetex Inc., Irvine, CA, USA), phospho-H2A.X, phospho-p53, caspase-9, caspase-3, mitogen-activated protein kinase kinases (MEK)1/2, phospho-MEK, phospho-extracellular regulated kinase (ERK), stress-activated ERK kinase (SEK)1, phospho-SEK, phospho-JNK, phospho-p38, c-Fos, c-Jun, phospho-c-Jun (Cell Signaling Technology, Danvers, MA, USA), B-cell lymphoma protein (Bcl)-2, Bcl-2 associated X (Bax), ERK2 (Santa Cruz Biotechnology, Dallas, TX, USA), IL-1β, matrix metalloproteinase (MMP)-2, MMP-9 (Abcam, Cambridge, MA, USA), p53, IL-6 (Invitrogen), MMP-1 (Cusabio, Houston, TX, USA).

### 2.11. Cell Cycle Analysis

We performed a cell cycle analysis to evaluate the effect of 3-BDB on PM_2.5_-induced cell cycle arrest. Cells were seeded into a 6-well plate, treated with 30 μΜ of 3-BDB for 1 h, and then with 50 μg/mL PM_2.5_ for 24 h. Propidium iodide and RNase A (1:1000) were used to bind to cellular DNA. This analysis was performed using a flow cytometer.

### 2.12. Hoechst 33342 Staining

We utilized Hoechst 33342 (BIOMOL GmbH, Hamburg, Germany) to visualize the protection of PM_2.5_-induced nuclei degradation by 3-BDB. Cells were seeded into a 60 mm culture dish and treated with 30 μΜ of 3-BDB for 1 h, followed by 50 μg/mL PM_2.5_ for 24 h. Then, the cells were immersed in a medium with Hoechst 33342, a DNA-specific fluorescent dye (10 µM) for 15 min. Stained cells were visualized under a fluorescence microscope (Olympus, Tokyo, Japan).

### 2.13. β-Galactosidase Staining Activity

We used a cellular senescence detection kit (SPiDER-β-Gal, Dojindo Laboratories, Kumamoto, Japan) to detect the expression of the senescence-associated enzyme, β-galactosidase (SA-β-gal) [29]. Cells were seeded into chamber slides and treated with 30 μΜ of 3-BDB for 1 h, followed by PM_2.5_ for another 24 h. After washing the chamber slides, the cells were stained with SPiDER-β-Gal solution and viewed under a confocal microscope.

### 2.14. Statistical Analysis

We performed statistical analyses among multiple groups by analyzing variance and Tukey’s tests using Systat 3.5 software (Systat Software Inc., San Jose, CA, USA). All data are displayed as mean ± standard error. The *p*-values less than 0.05 were considered statistically significant.

## 3. Results

### 3.1. Antioxidant Effect of 3-BDB against PM_2.5_-Induced Intracellular ROS and Lipid Peroxidation

We confirmed the ROS scavenging effect of 3-BDB at 10, 20, or 30 μM, and NAC (1 mM) was used as a positive control induced by PM_2.5_ (Figure 1a). The results proved that 30 μΜ of 3-BDB, such as NAC, significantly prevented cells from PM_2.5_-induced ROS. Next, confocal images confirmed that cells treated with 30 μΜ of 3-BDB contained lower ROS than the PM_2.5_ group (Figure 1b). Thus, in the following trials, we used 30 μΜ as the optimal concentration of 3-BDB. Furthermore, to investigate ROS-induced lipid peroxidation, cells were subjected to 3-BDB and/or PM_2.5_. Findings revealed that PM_2.5_ caused lipid damage, whereas 3-BDB had an antagonistic effect (Figure 1c).

### 3.2. Preventive Effect of 3-BDB against PM_2.5_-Induced Mitochondrial Dysfunction

As shown in Figure 2a, PM_2.5_ increased mitochondrial calcium level, whereas the treatment with 3-BDB and PM_2.5_ led to a decreased calcium level than PM_2.5_. The mitochondrial membrane potential was analyzed to further assess mitochondrial dysfunction. The mitochondrial membrane potential depolarized by PM_2.5_ was reversed after treatment with 3-BDB, as displayed by results from flow cytometry and confocal microscopic images (Figure 2b,c).

### 3.3. Inhibitory Effect of 3-BDB against PM_2.5_-Induced DNA Damage

Cells treated with 3-BDB and PM_2.5_ possessed a lower level of 8-oxoG than those in the PM_2.5_-treated group. This implies that 3-BDB inhibited PM_2.5_-induced DNA oxidation (Figure 3a). Similar results confirmed that 3-BDB protected PM_2.5_-induced DNA damage in the comet assay because 3-BDB reduced the DNA tail length induced by PM_2.5_ in cells (Figure 3b). In addition, we detected phospho-H2A.X histone, a known indicator of DNA double-strand break [30], and a significant increase in phosphorylation of H2A.X in the PM_2.5_ group in vitro and in vivo was observed; however, in the 3-BDB and PM_2.5_ treatment group, a significant decrease in phospho-H2A.X was observed (Figure 3c,d). ROS and ROS-induced DNA damage activate p53, a known tumor suppressor [31]. There was an increase in the phosphorylation of p53 in the PM_2.5_ group; however, after treatment with 3-BDB, the level of activated p53 was lower in vitro and in vivo (Figure 3c,d). Furthermore, p53 controls the fate of cells when subjected to DNA damage, probably by arresting cell cycle progression [32,33]. Notably, the cell cycle analysis revealed that PM_2.5_ arrested the cell cycle at the G_1_ phase; however, 3-BDB attenuated it (Figure 3e).

### 3.4. Anti-Apoptotic Effect of 3-BDB against PM_2.5_-Induced Apoptosis

PM_2.5_ decreased the anti-apoptotic protein, Bcl-2, and increased the pro-apoptotic protein, Bax; however, 3-BDB reversed these effects in vitro and in vivo (Figure 4a,b). 3-BDB also reversed PM_2.5_-activated caspase-9 and caspase-3, the main markers of apoptosis-mediated cell death, in vitro and in vivo (Figure 4c,d). Nuclei integrity was visualized through Hoechst 33342 staining. Notably, PM_2.5_ increased apoptotic bodies, but 3-BDB significantly decreased the number of apoptotic bodies (Figure 4e).

### 3.5. Inactivating Effect of 3-BDB against PM_2.5_-Induced Activator Protein (AP)-1 via Mitogen- Activated Protein Kinase (MAPK) Signaling Pathway

AP-1 transcription factor is associated with MAPK-induced apoptosis [34]. Thus, we checked the expression levels of MAPK-related proteins, MEK, ERK, SEK, JNK, and p38. PM_2.5_ induced a high level of activated MEK1/2, ERK1/2, SEK1, JNK, and p38, which were reversed by treatment with 3-BDB (Figure 5a). Besides, the transcription factor AP-1 (c-Jun and c-Fos) was also activated by PM_2.5_, but it decreased in the 3-BDB- and PM_2.5_-treated groups (Figure 5b).

### 3.6. Antagonizing Effect of 3-BDB against PM_2.5_-Induced Senescence

We mainly examined pro-inflammatory cytokines, senescence-related proteins, and markers. The levels of the pro-inflammatory cytokines IL-1β (Figure 6a) and IL-6 (Figure 6b) were induced in PM_2.5_-exposed cells, while 3-BDB reduced levels of IL-1β and IL-6 (Figure 6a,b). Moreover, IL-1β and IL-6 protein levels were also higher in the PM_2.5_ group than in the 3-BDB + PM_2.5_-treated group in vitro and in vivo (Figure 6c,d). PM_2.5_ also induced MMP-1, MMP-2, and MMP-9 expression; however, these effects were reversed by 3-BDB both in vitro and in vivo (Figure 6e,f). Finally, we examined senescent cells through staining with SA-β-gal [13]. The results showed that PM_2.5_ generates higher fluorescence than the control group; however, 3-BDB notably inhibited the fluorescence intensity induced by PM_2.5_ (Figure 6g).

## 4. Discussion

Particulate matter (PM_2.5_) is a strong matter of interest nowadays, as research is ongoing to determine its effect on human bodies. PM_2.5_ possesses a different degree of toxicity, confirming that diesel engine combustion was more severe than biomass burning in the same mass [35]. One of the main organic components (polycyclic aromatic hydrocarbons, PAHs) from engine exhaust generates ROS, resulting in DNA damage [36]. Therefore, in this study, we used the recommended PM_2.5_, mainly from diesel particulate matter, including PAHs and nitro-PAHs. Although, the skin can deal with different sources of ROS through a specific antioxidant mechanism; PM_2.5_ overloaded it with high concentrations of ROS [37]. For ROS scavenging, we focused on the antioxidant compound 3-BDB, obtained from red algae, as it possesses a strong ability to protect against oxidative stress-related cell damage, including inflammation [21,22,23,24]. We previously showed that there was no cytotoxicity of 3-BDB at concentrations ranging from 10–30 μΜ in human HaCaT keratinocytes, and 3-BDB inhibited UVB-caused oxidative stress at 30 μΜ concentration [38]. In addition, PM_2.5_ generated ROS [12,13,14,15]. Moreover, lipid peroxidation is vital for initiating the process of cell damage because lipids are prime targets of free radicals [39]. As shown in Figure 1, 3-BDB pretreatment alleviates both PM_2.5_-induced cellular ROS generation and lipid peroxidation.

ROS are mainly generated from mitochondria, which are closely related to proton leaks [40]. However, oxidative stress via excessive ROS induces mitochondrial dysfunction associated with DNA damage via depolarization of mitochondrial membrane potential [41]. Mitochondrial calcium homeostasis is vital for proper mitochondrial function, but Ca^2+^ can also trigger the mitochondrial apoptosis pathway [42]. The increased ROS by PM_2.5_ decreases mitochondrial action potential, causing apoptosis [12]. Furthermore, changes in the mitochondria are necessary for the senescence phenotype [43]. Thus, we examined mitochondrial calcium levels and membrane potential. Since 3-BDB inhibits ROS formation, it exerts protective effects on the mitochondria from calcium- and membrane-depolarization-induced dysfunction.

Oxidative stress leads to DNA oxidation and mutation, cancer, and senescence [29]. PM_2.5_-induced oxidative stress causes DNA damage, which leads to cell cycle arrest in skin cells [44]. In this study, we noted less DNA damage in the 3-BDB and PM_2.5_ groups than in the PM_2.5_ group (Figure 3). One way to induce senescence is by activating the tumor suppressor, p53, which can be activated by oxidative stress or DNA damage [33]. p53 is a key factor for determining cell fate; under stress, it can maintain G_1_ arrest to accelerate aging [31]. Our results were in agreement with the above explanation and demonstrated that PM_2.5_ stimulated the activation of p53 and caused G_1_ arrest; however, these effects were reversed by treatment with 3-BDB.

Our previous studies showed that PM_2.5_ leads to cell apoptosis via ROS [12,14,37,45], and from the cell cycle analysis, PM_2.5_ could induce aging and apoptosis through cell cycle arrest. Moreover, mitochondrial dysfunction is involved in apoptosis and aging [45,46]. We examined the effect of 3-BDB on PM_2.5_-induced apoptosis. The results shown in Figure 4 show that 3-BDB reduces apoptotic bodies, as it inhibits ROS generation. Excessive ROS induced MMPs via the MAPK-transcription factor AP-1 signaling pathway [47,48]. Furthermore, ROS increases the secretion of pro-inflammatory cytokines, which are secreted at a high level in most senescent cells [49].

DNA damage has been regarded as an activator of the senescence-associated secretory phenotype (SASP), which is related to cell cycle arrest [50]. Two key SASP factors, IL-1β and IL-6, were detected at high levels in senescent cells [33]. Furthermore, we previously demonstrated that PM_2.5_ increases levels of IL-1β and IL-6 [14]. MAPK induces the phosphorylation of NF-κB that promotes the secretion of pro-inflammatory cytokines and regulates AP-1 [51]. Collagen degradation is probably related to the formation of MMPs, especially in the epidermis and dermis [52]. Our previous study showed that PM_2.5_ induces the production of MMPs (MMP-1, MMP-2, and MMP-9) and eventually induces cell senescence [13]. In the present study, we observed that PM_2.5_ activated the MAPK signaling pathway and transcription factors (Figure 5), followed by the secretion of pro-inflammatory cytokines and MMPs; however, 3-BDB relieved cells from the stress condition induced by PM_2.5_ (Figure 6). The senescence marker β-galactosidase is present in aged cells [29]. As shown in Figure 6e, β-galactosidase activity cells were stimulated by PM_2.5_, but they were decreased via pretreatment with 3-BDB.

## 5. Conclusions

In summary, the inhibition of ROS generation by 3-BDB in human HaCaT keratinocytes and hairless mice reduces mitochondrial dysfunction and DNA damage response, which inhibits activation of the tumor suppressor p53 and cell cycle arrest. In addition, 3-BDB affects the inhibition of the MAPK signaling pathway and its regulated transcription factor, AP-1, reversing the formation of pro-inflammatory cytokines and MMPs, thereby inhibiting PM_2.5_-induced senescence by 3-BDB (Figure 7). Notably, 3-BDB had a protective effect against PM_2.5_-induced cellular damage and could be used as a preventive agent against air pollution-triggered skin damage.

## Figures and Tables

**Figure 1 antioxidants-12-01307-f001:**
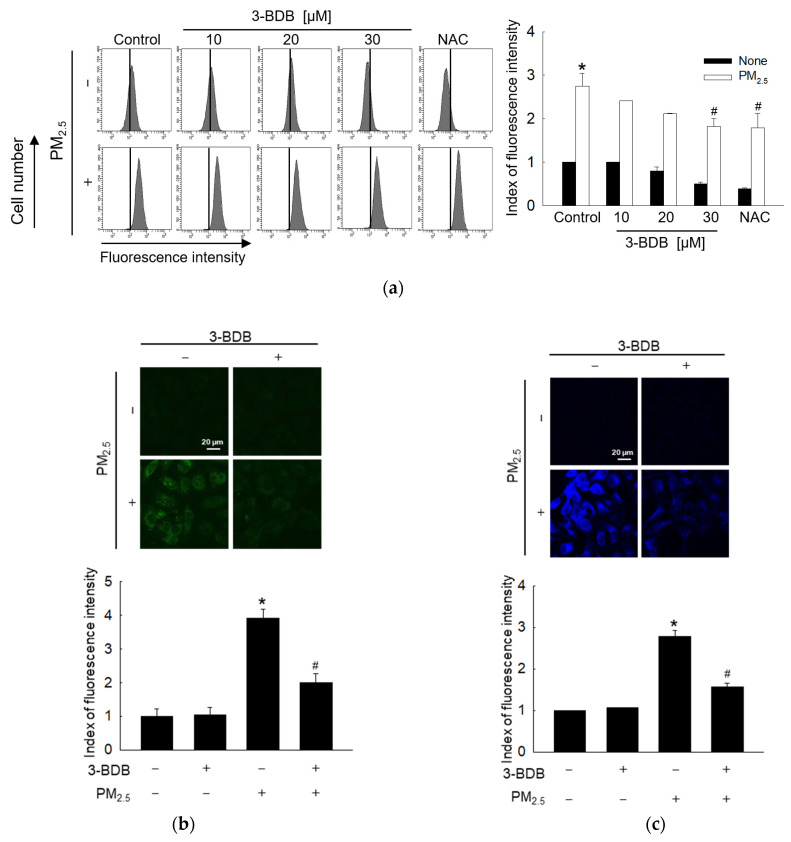
Inhibition of PM_2.5_-induced ROS generation and lipid peroxidation were performed by 3-BDB in keratinocytes. (**a**) Cells were added to 10, 20, and 30 μΜ of 3-BDB or 1 mM of N-acetyl cysteine (NAC) for 1 h and then exposed to 50 μg/mL of PM_2.5_ for 30 min. ROS were measured by a flow cytometer after H_2_DCFDA staining. (**b**) Depletion of PM_2.5_-induced ROS by 30 μM of 3-BDB was visualized by a confocal microscope after H_2_DCFDA staining. (**c**) Prevention of PM_2.5_-induced lipid peroxidation analysis by 3-BDB was performed using a confocal microscope after DPPP staining. (**a**–**c**) * *p* < 0.05 and ^#^
*p* < 0.05 compared to control cells and PM_2.5_-exposed cells, respectively.

**Figure 2 antioxidants-12-01307-f002:**
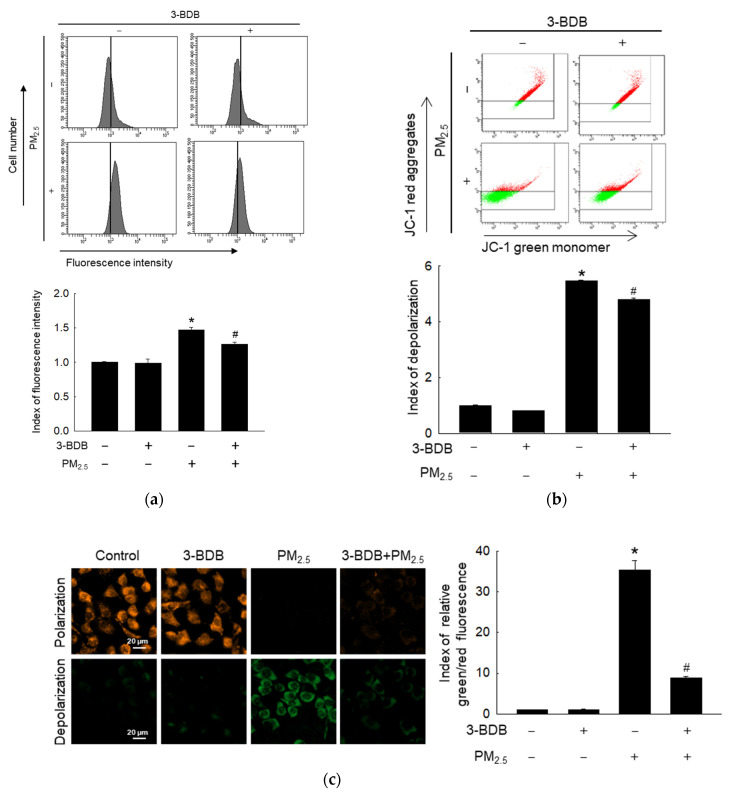
Prevention of PM_2.5_-induced mitochondrial dysfunction was performed by 3-BDB in keratinocytes. Cells were treated with 30 μΜ of 3-BDB for 1 h and then exposed to 50 μg/mL of PM_2.5_ for 24 h. (**a**) Rhod-2 AM was used to detect the mitochondrial calcium. (**b**,**c**) The mitochondrial membrane potential was obtained by (**b**) flow cytometry and (**c**) confocal microscopy by JC-1 staining. (**a**–**c**) * *p* < 0.05 and ^#^
*p* < 0.05 compared to control cells and PM_2.5_-exposed cells, respectively.

**Figure 3 antioxidants-12-01307-f003:**
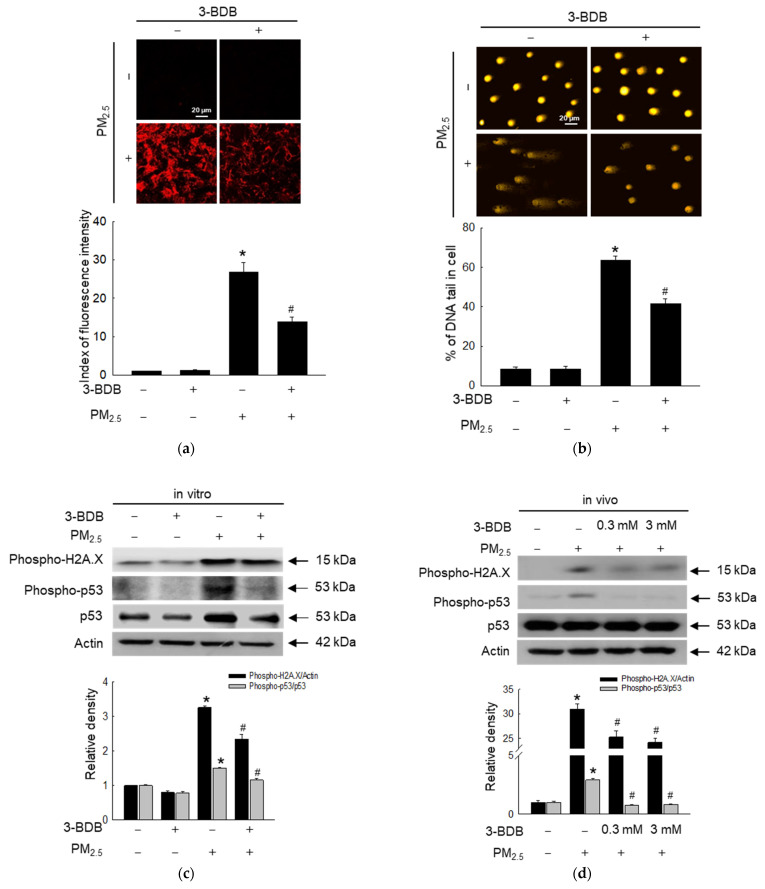
Reversibility of PM_2.5_-induced DNA damage and cell cycle arrest was performed by 3-BDB. Cells were treated with 30 μΜ of 3-BDB for 1 h and then exposed to 50 μg/mL of PM_2.5_ for 24 h. (**a**) Avidin-TRITC conjugate was used to detect the 8-oxoG. (**b**) A Comet assay was performed to analyze DNA damage. (**c**,**d**) The proteins were obtained from both (**c**) cells and (**d**) tissues, and phospho-H2A.X, phospho-p53, and p53 were examined by Western blot. (**e**) The checkpoint of the G_1_ phase was measured by flow cytometry. (**a**–**e**) * *p* < 0.05 and ^#^
*p* < 0.05 compared to control groups and PM_2.5_-exposed groups, respectively.

**Figure 4 antioxidants-12-01307-f004:**
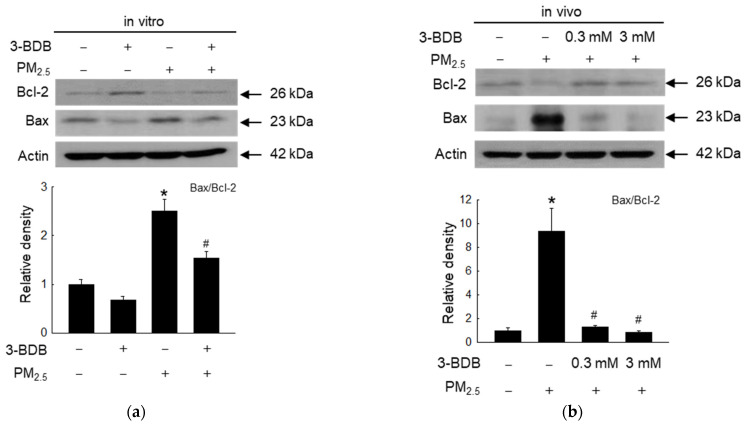
Reduction in PM_2.5_-induced cell apoptosis was by 3-BDB in vitro and in vivo. Cells were treated with 30 μΜ of 3-BDB for 1 h and then exposed to 50 μg/mL of PM_2.5_ for 24 h. Mice skin was treated with 3-BDB and PM_2.5_ according to the animal experiment in Materials and Methods. (**a**–**d**) The proteins were isolated from (**a**,**c**) cells, (**b**,**d**) tissue, and Bcl-2, Bax, and cleaved caspase-9, and cleaved caspase-3 were detected by Western blot. (**e**) The apoptotic bodies were counted by using Hoechst 33342 staining. The arrows indicate the apoptotic bodies. (**a**–**e**) * *p* < 0.05 and ^#^
*p* < 0.05 compared to control groups and PM_2.5_-exposed groups, respectively.

**Figure 5 antioxidants-12-01307-f005:**
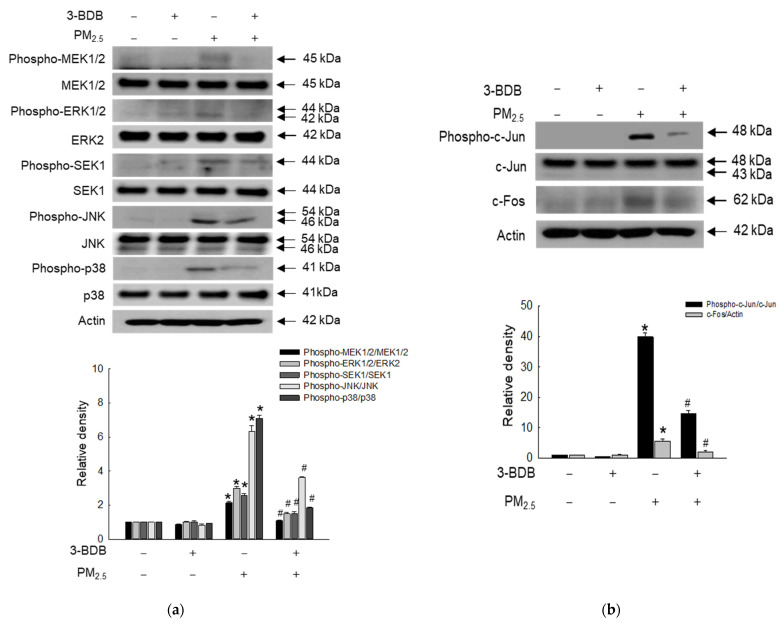
Inactivation of PM_2.5_-induced MAPK signaling pathway, the transcription factor AP-1 was performed by 3-BDB. (**a**,**b**) Cells were treated with 30 μΜ of 3-BDB for 1 h, and then the cells were stimulated by PM_2.5_ for 24 h, the proteins were separated from cells, and (**a**) phospho-MEK1/2, MEK1/2, phospho-ERK1/2, ERK2, phospho-SEK1, SEK1, phospho-JNK, JNK, phospho-p38, p38, as well as (**b**) c-Fos, phospho-c-Jun, and c-Jun expressions were detected by Western blot. (**a**,**b**) * *p* < 0.05 and ^#^
*p* < 0.05 compared to control cells and PM_2.5_-exposed cells, respectively.

**Figure 6 antioxidants-12-01307-f006:**
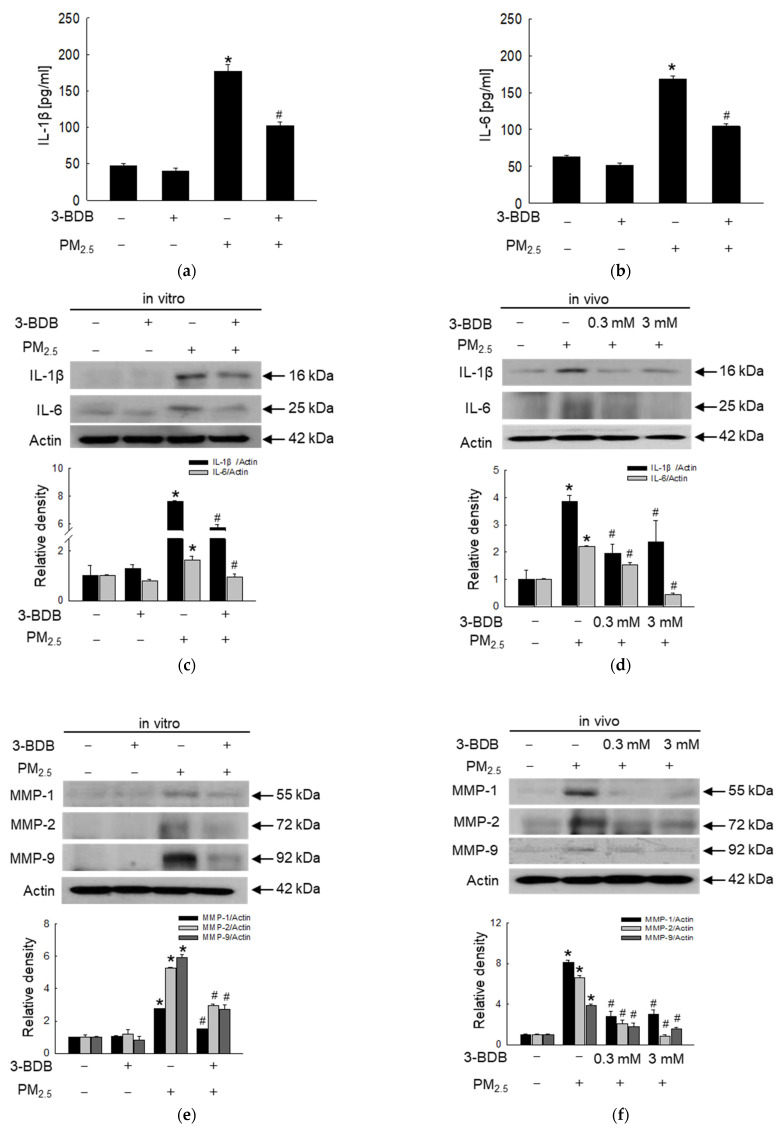
Inhibition of PM_2.5_-induced pro-inflammatory cytokines and matrix metalloproteinases was performed by 3-BDB in vitro and in vivo. Cells were treated with 30 μΜ of 3-BDB for 1 h and then were exposed to 50 μg/mL of PM_2.5_ for 24 h. Mice skin was treated with 3-BDB and PM_2.5_ according to the animal experiment in Materials and Methods. (**a**,**b**) IL-1β and IL-6 concentrations in HaCaT cells were assessed using a human IL-1β and IL-6 ELISA kits, respectively. (**c**–**f**) From the proteins of (**c**,**e**) cells and (**d**,**f**) tissues, IL-1β, IL-6, MMP-1, MMP-2, and MMP-9 were examined by Western blot. (**g**) Senescence cells were available to visualize under a confocal microscope. (**a**–**g**) * *p* < 0.05 and ^#^
*p* < 0.05 compared to control groups and PM_2.5_-exposed groups, respectively.

**Figure 7 antioxidants-12-01307-f007:**
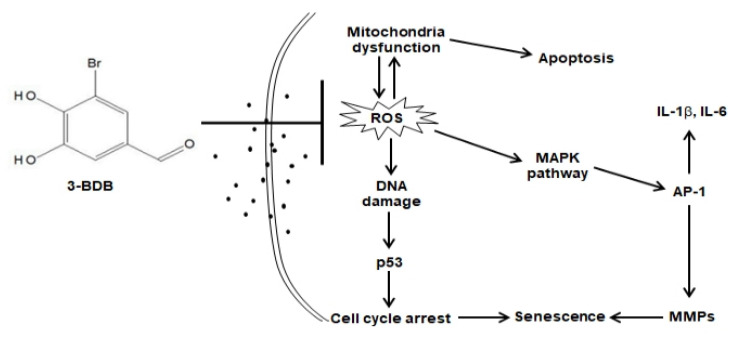
The schematic diagram for the protective effect of 3-BDB induced by PM_2.5_ was exhibited. 3-BDB inhibited ROS generation induced by PM_2.5_, which caused macromolecular damage, cell cycle arrest, and apoptosis. In addition, 3-BDB inhibited inflammatory cytokines release and MMPs expression through the MAPK signaling pathway, thus alleviating cell senescence through a complex intracellular mechanism.

## Data Availability

The data presented in this study are available upon request from the corresponding author.

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
