# Peer review of "3-Bromo-4,5-dihydroxybenzaldehyde Protects Keratinocytes from Particulate Matter 2.5-Induced Damages"

_antioxidants, 2023, doi:10.3390/antiox12061307_

Round 1

Reviewer 1 Report

Particle Matters 2.5 (PM2.5) are continuously discharged into air and causing severe health problems such as organizing pneumonia and/or lung cancer. In this manuscript, Zhen AX et al demonstrates that 3-bromo-4,5-dihydroxybenzaldehyde (3-BDB) attenuated PM2.5-induced ROS and inflammatory responses. This manuscript is well-organized; however, following points should be clarified.

Major points

#1: How did authors detect 8-oxoguanine in figure 3A? It may hard to detect by avidin-TRITC conjugates.

#2: In figure 7, authors demonstrates that MAPK pathway are important in 3-BDB-induced attenuation. In figure 5, phosphorylated levels of JNK, p38, c-Fos, c-Jun are shown to be modulated significantly by PM2.5 with 3-BDB. Please show the protein levels of non-phosphorylated form.

#3: Do you keep supernatant (medium) of PM2.5-treated cells? IL-1beta and IL-6 may be released into media after PM2.5 exposure.

Minor points

##1: In figure 1B, is this stained by H2DCFDA?

##2: Please insert molecular weight marker in western blot.

##3: It may be nice to show p53 immunoblot in figure 3C,D.

English is fine.

Author Response

Comments and Suggestions for Authors

Particle Matters 2.5 (PM2.5) are continuously discharged into air and causing severe health problems such as organizing pneumonia and/or lung cancer. In this manuscript, Zhen AX et al demonstrates that 3-bromo-4,5-dihydroxybenzaldehyde (3-BDB) attenuated PM2.5-induced ROS and inflammatory responses. This manuscript is well-organized; however, following points should be clarified.

Major points

#1: How did authors detect 8-oxoguanine in figure 3A? It may hard to detect by avidin-TRITC conjugates.

 Response: 8-Oxoguanine (8-oxoG) is the most significant biomarker for oxidative DNA damage (Chiorcea-Paquim 2022. To detect 8-oxoG levels, we used the avidin-tetramethylrhodamine isothiocyanate (TRITC) conjugate fluorescent dye (Sigma-Aldrich Co.), which has an affinity to 8-oxoG (Conner et al., 2006). Harvest cells in the chamber slide were treated with 3-BDB (30 μΜ) and PM2.5 for another 24 h. Then, cells were stained by avidin-TRITC conjugate, and the fluorescence intensity of stained cells were estimated using a software program of image J under the confocal microscope (Piao et al., 2018). It was described in 2.7. Detection of 8-oxoguanine (8-oxoG).

Chiorcea-Paquim, A.M. 8-oxoguanine and 8-oxodeoxyguanosine biomarkers of oxidative DNA aamage: a Review on HPLC-ECD determination. Molecules 2022, 27, 1620.

Conners, R.; Hooley, E.; Clarke, A.R.; Thomas, S.; Brady, R.L. Recognition of oxidatively modified bases within the biotin-binding site of avidin. J. Mol. Biol. 2006, 357, 263-274.

Piao, M.J.; Ahn, M.J.; Kang, K.A.; Ryu, Y.S.; Hyun, Y.; Shilnikova, K.; Zhen, A.X.; Jeong, J.W.; Choi, Y.H.; Kang, H.K.; et al. Particulate matter 2.5 damages skin cells by inducing oxidative stress, subcellular organelle dysfunction, and apoptosis. Arch. Toxicol. 2018, 92, 2077-2091.

#2: In figure 7, authors demonstrates that MAPK pathway are important in 3-BDB-induced attenuation. In figure 5, phosphorylated levels of JNK, p38, c-Fos, c-Jun are shown to be modulated significantly by PM2.5 with 3-BDB. Please show the protein levels of non-phosphorylated form.

Response: We measured protein levels of non-phosphorylated MEK, ERK, SEK, JNK, p38, and c-Jun. The results are provided in Fig. 5a and 5b.

#3: Do you keep supernatant (medium) of PM2.5-treated cells? IL-1beta and IL-6 may be released into media after PM2.5 exposure.

 Response: As reviewer’s comment, we measured the concentration of IL-1beta and IL-6 in the media after PM2.5 exposure in vitro using the ELISA kit. The results are shown in Fig. 6a and 6b. Also, the detection of IL-1beta and IL-6 using ELISA kits was added in 2.9. Detection of IL-1β and IL-6.

Minor points

##1: In figure 1B, is this stained by H2DCFDA?

 Response: The cellular ROS level in Figure 1b was determined by H2DCFDA staining. According to the reviewer’s valuable suggestion, the legend of Figure 1b was revised as follows “(b) Depletion of PM2.5-induced ROS by 3-BDB (30 μM) was visualized by a confocal microscope after H2DCFDA staining”.

##2: Please insert molecular weight marker in western blot.

 Response: The molecular weight markers in western blot were provided in western blot on all figures according to the reviewer’s comment.

##3: It may be nice to show p53 immunoblot in figure 3C, D.

Response: The p53 immunoblot was provided in Figure 3c and 3d according to the reviwer’s comment.

Reviewer 2 Report

The study is a continuation of investigation on molecular mechanism of action 3-bromo-4,5-dihydroxybenzaldehyde (BDB) carried out by the Authors. The results are interesting; however, the manuscript needs some improvement.

1)      The abstract should be structured: background, purpose of the study; methods, results, conclusions (see MDPI template). There is no need to describe the previous papers here (lines 18-22).

2)      Introduction: the aim of the study should be clearly described. Furthermore, more information on 3-bromo-4,5-dihydroxybenzaldehyde should be added.

3)      Materials and methods: 2.1. “DMSO was used as vehicle control.” – pure DMSO? DMSO is cytotoxic. What was the final concentration DMSO in samples? What solvent was used to dilute samples? What solvent was used to dissolve 3-BDB? What was the final concentration?

4)      2.3. Experimental design is unclear. More detail should be added. Was mice skin treated with 3-BDB and PM2.5 simultaneously? How were the samples prepared for analysis?

5)      2.5. were 3-BDB and PM2.5 used simultaneously?

6)      2.6. Lack of information on treatment of cells with 3-BDB/ PM2.5

7)      Results: this section should focused only on presentation the results. For example lines 155-158, 176-180 … should be moved to Discussion. Please check the entire section.

8)      Figure 1a: “The human keratinocytes were subjected to PM2.5 exposure for 30 min..” – and when 3-BDB was added? Before or after exposure? 1 c “Prevention of PM2.5-induced lipid peroxidation (…) was performed by DPPP staining.” – not precise. Was visualized by DPPP staining?

9)      “Mitochondrial calcium homeostasis is vital in mitochondrial dysfunction, much of which triggers the mitochondrial apoptosis pathway” – unclear sentence. In dysfunction? I think that calcium homeostasis is vital in mitochondrial proper function.

10)   Figure 2: “mitochondrial dysfunction was performed by 3-BDB pretreatment in keratinocytes.” – lack of detail. What was the time of pretreatment?

11)   Figure 3 a: “Cells were stimulated by PM2.5 for 24 h” – only PM2.5? what about 3-BDB? Please check all figure legends.

12)   „Excessive ROS generation occurs matrix metallopeptidases (MMPs)…” – unlear expression. Reedit.

13)   Line 246: “Furthermore, ROS could help in the secretion of proinflammatory cytokines…“ – the word “help” suggests that it is a positive action.

14)   Line 253: “PM2.5 generates higher fluorescence” – higher than what?

15)   Figure 6 legend: should be “metalloproteinases”

16)   Discussion should be better.  Now, it is a repetition of information from the results section with short explanation why such factors was investigated. Discussion in the relation with the previous published Authors findings could be interesting.

17)   Line 283: „It has been shown that excessive ROS stimulates the cell cycle as it arrests oxidative stress-induced DNA damage” – ROS arrests oxidative stress-induced DNA damage? excessive ROS can contribute to oxidative stress, which can lead to DNA damage

Author Response

Comments and Suggestions for Authors

The study is a continuation of investigation on molecular mechanism of action 3-bromo-4,5-dihydroxybenzaldehyde (BDB) carried out by the Authors. The results are interesting; however, the manuscript needs some improvement.

1) The abstract should be structured: background, purpose of the study; methods, results, conclusions (see MDPI template). There is no need to describe the previous papers here (lines 18-22).

Response: We have edited the abstract section according to the reviewer’s suggestions.

2) Introduction: the aim of the study should be clearly described. Furthermore, more information on 3-bromo-4,5-dihydroxybenzaldehyde should be added.

Response: The introduction section was revised according to the reviwer’s comments as follows; “3-bromo-4,5-dihydroxybenzaldehyde (3-BDB), a natural marine compound from red algae (Rhodomela confervoides, Polysiphonia morrowii, and Polysiphonia urceolata), possesses free radical scavenging, anticancer, and antibacterial properties. [Cho et al., 2019; Jayasinghe et al., 2022; Kim et al., 2011]”. “We aimed at elucidating the effect of 3-BDB on PM2.5-induced ROS generation, macro molecular damages, apoptosis, and senescence in skin cells in vitro and in vivo”.

Cho, S.H.; Heo, S.J.; Yang, H.W.; Ko, E.Y.; Jung, M.S.; Cha, S.H.; Ahn, G.; Jeon, Y.J.; Kim, K.N. Protective effect of 3-Bromo-4,5-dihydroxybenzaldehyde from Polysiphonia morrowii harvey against hydrogen peroxide-induced oxidative stress in vitro and in vivo. J. Microbiol. Biotechnol. 2019, 29, 1193-1203.

Kim, S.Y.; Kim, S.R.; Oh, M.J.; Jung, S.J.; Kang, S.Y. In vitro antiviral activity of red alga, Polysiphonia morrowii extract and its bromophenols against fish pathogenic infectious hematopoietic necrosis virus and infectious pancreatic necrosis virus. J. Microbiol. 2011, 49, 102–106.

Jayasinghe, A.M.K.; Han, E.J.; Kirindage, K.G.I.S.; Fernando, I.P.S.; Kim, E.A.; Kim, J.; Jung, K.; Kim, K.N.; Heo, S.J.; Ahn, G. 3-bromo-4,5-dihydroxybenzaldehyde isolated from Polysiphonia morrowii suppresses TNF-α/IFN-γ-stimulated inflammation and deterioration of skin barrier in HaCaT keratinocytes. Mar. Drugs 2022, 20, 563.

3) Materials and methods: 2.1. “DMSO was used as vehicle control.” – pure DMSO? DMSO is cytotoxic. What was the final concentration DMSO in samples? What solvent was used to dilute samples? What solvent was used to dissolve 3-BDB? What was the final concentration?

Response: 3-BDB and PM2.5 were dissolved in dimethyl sulfoxide (DMSO), and the concentration of DMSO in the cell medium during treatment was maintained at < 0.1%. Some studies show that < 0.5% final DMSO concentration was not cytotoxic (Borja-Martínez et al., 2020). It is described in 2.1. Sample Preparation.

Borja-Martínez, M.; Lozano-Sánchez, J.; Borrás-Linares, I.; Pedreño, M.A.; Sabater-Jara, A.B. Revalorization of broccoli by-products for cosmetic uses using supercritical fluid extraction. Antioxidants 2020, 9, 1195.

4)  2.3. Experimental design is unclear. More detail should be added. Was mice skin treated with 3-BDB and PM2.5 simultaneously? How were the samples prepared for analysis?

Response: We added that in 2.3. Animal Experiment as pointed out by the reviewer.

“We used HR-1 hairless male mice (OrientBio, Kyungki-do, Republic of Korea) for in vivo experiments following the guidelines of the Jeju National University (Jeju, Republic of Korea) (permit number: 2017-0026). Moreover, mice were divided into four groups (n = 4 per group): phosphate-buffered saline (PBS; normal), PM2.5 (100 µg/mL), 3-BDB (0.3 mM) + PM2.5, and 3-BDB (3 mM) + PM2.5. The dorsal surface of the skin of the mice was exposed to 3-BDB (0.3 or 3 mM) 30 min before exposing them to PM2.5. Then, they were covered with the nonwoven polyethylene pad (over a 1 cm2 area), which dispersed PM2.5 daily for seven days consecutively. Finally, on day 7, the skin tissues were dissected for western blot analysis [12].”

5) 2.5. were 3-BDB and PM2.5 used simultaneously?

Response: Cells were seeded into chamber slides, treated with 3-BDB (30 μΜ) for 1 h, and exposed to PM2.5 (50 μg/mL) for another 24 h. It was described in 2.5. Lipid Peroxidation Assay.

6) 2.6. Lack of information on treatment of cells with 3-BDB/ PM2.5

Response: Cells were treated with 3-BDB (30 μΜ) for 1 h, and exposed to PM2.5 (50 μg/mL) for another 24 h. It was described in 2.6. Analysis of Mitochondria Function.

7) Results: this section should focused only on presentation the results. For example lines 155-158, 176-180 … should be moved to Discussion. Please check the entire section.

Response: The Results and Discussion sections were edited according to the reviewer’s comment.

8) Figure 1a: “The human keratinocytes were subjected to PM2.5 exposure for 30 min..” – and when 3-BDB was added? Before or after exposure? 1 c “Prevention of PM2.5-induced lipid peroxidation (…) was performed by DPPP staining.” – not precise. Was visualized by DPPP staining?

Response: The cells were pretreated with 30 μΜ of 3-BDB for 1 h, and then exposed to 50 μg/mL of PM2.5 for 30 min. It was revised in the legend of Figure 1a. Prevention of PM2.5-induced lipid peroxidation analysis by 3-BDB was performed using a confocal microscope after DPPP staining. It was revised in the legend of Figure 1c.

9) “Mitochondrial calcium homeostasis is vital in mitochondrial dysfunction, much of which triggers the mitochondrial apoptosis pathway” – unclear sentence. In dysfunction? I think that calcium homeostasis is vital in mitochondrial proper function.

Response: Mitochondrial calcium homeostasis is vital in mitochondrial proper function; however, calcium can also trigger the mitochondrial apoptosis pathway. It was revised in the Discussion.

10) Figure 2: “mitochondrial dysfunction was performed by 3-BDB pretreatment in keratinocytes.” – lack of detail. What was the time of pretreatment?

Response: Cells were treated with 30 μΜ of 3-BDB for 1 h, and then exposed to 50 μg/mL of PM2.5 for 24 h. It was revised in the legend of Figure 2.

11)  Figure 3 a: “Cells were stimulated by PM2.5 for 24 h” – only PM2.5? what about 3-BDB? Please check all figure legends.

Response: Cells were treated with 30 μΜ of 3-BDB for 1 h, and then exposed to 50 μg/mL of PM2.5 for 24 h. It was revised in the legend of Figure 3.

12)  „Excessive ROS generation occurs matrix metallopeptidases (MMPs)…” – unlear expression. Reedit.

Response: Excessive ROS induced matrix metalloproteinases (MMPs) via MAPK- transcription factor AP-1 signaling pathway. It was revised in the Discussion section.

13)  Line 246: “Furthermore, ROS could help in the secretion of proinflammatory cytokines…“ – the word “help” suggests that it is a positive action.

Response: We changed “help” to “induce” as pointed out in the reviewer’s comment.

14)  Line 253: “PM2.5 generates higher fluorescence” – higher than what?

Response: The PM2.5 group generates higher fluorescence than control group. It was revised in 3.6. Antagonizing Effect of 3-BDB against PM2.5-Induced Senescence.

15)  Figure 6 legend: should be “metalloproteinases”

Response: This was revised according to the reviewer’s comment.

16)  Discussion should be better.  Now, it is a repetition of information from the results section with short explanation why such factors was investigated. Discussion in the relation with the previous published Authors findings could be interesting.

Response: The Discussion section was edited, as pointed out in the reviewer’s comment.

17)  Line 283: „It has been shown that excessive ROS stimulates the cell cycle as it arrests oxidative stress-induced DNA damage” – ROS arrests oxidative stress-induced DNA damage? excessive ROS can contribute to oxidative stress, which can lead to DNA damage

Response: PM2.5-induced oxidative stress causes DNA damage, leading to cell cycle arrest in skin cells. It was revised in Discussion section.

Round 2

Reviewer 2 Report

The manuscript has been improved according to my suggestions and it can be published in current form.